# Antibacterial and Analgesic Properties of Beta-Caryophyllene in a Murine Urinary Tract Infection Model

**DOI:** 10.3390/molecules28104144

**Published:** 2023-05-17

**Authors:** Kayle Dickson, Cassidy Scott, Hannah White, Juan Zhou, Melanie Kelly, Christian Lehmann

**Affiliations:** 1Department of Microbiology and Immunology, Dalhousie University, Halifax, NS B3H 4R2, Canada; 2Department of Pharmacology, Dalhousie University, Halifax, NS B3H 4R2, Canada; cassidy.scott@dal.ca (C.S.); hannahwhite@dal.ca (H.W.);; 3Department of Anesthesiology, Pain Management and Perioperative Medicine, Dalhousie University, Halifax, NS B3H 4R2, Canada; juan.zhou@dal.ca; 4Department of Physiology and Biophysics, Dalhousie University, Halifax, NS B3H 4R2, Canada

**Keywords:** cannabis, terpenes, pain, inflammation, intravital microscopy

## Abstract

Beta-caryophyllene has demonstrated anti-inflammatory effects in a variety of conditions, including interstitial cystitis. These effects are mediated primarily via the activation of the cannabinoid type 2 receptor. Additional antibacterial properties have recently been suggested, leading to our investigation of the effects of beta-caryophyllene in a murine model of urinary tract infection (UTI). Female BALB/c mice were intravesically inoculated with uropathogenic *Escherichia coli* CFT073. The mice received either beta-caryophyllene, antibiotic treatment using fosfomycin, or combination therapy. After 6, 24, or 72 h, the mice were evaluated for bacterial burden in the bladder and changes in pain and behavioral responses using von Frey esthesiometry. In the 24 h model, the anti-inflammatory effects of beta-caryophyllene were also assessed using intravital microscopy. The mice established a robust UTI by 24 h. Altered behavioral responses persisted 72 h post infection. Treatment with beta-caryophyllene resulted in a significant reduction in the bacterial burden in urine and bladder tissues 24 h post UTI induction and significant improvements in behavioral responses and intravital microscopy parameters, representing reduced inflammation in the bladder. This study demonstrates the utility of beta-caryophyllene as a new adjunct therapy for the management of UTI.

## 1. Introduction

Urinary tract infections (UTIs) are a common type of infection involving the urethra, bladder, and/or kidneys which affect more than 400 million people annually [1,2]. Premenopausal females are disproportionately affected, experiencing UTIs 40 times as frequently as males of the same age [3]. Symptoms of a UTI affecting the lower tract (i.e., bladder) include increased urinary frequency, increased urinary urgency, pain, and blood in the urine. While not generally life-threatening, UTI symptoms are uncomfortable and decrease overall quality of life. If left untreated, lower UTIs may spread into the kidneys, resulting in systemic symptoms including fever and flank pain. Further progression of infection may result in serious sequelae, including sepsis and death. While most infections are not life-threatening and resolve with antibiotic therapy, UTIs cost the healthcare system billions of dollars each year [1]. This high cost is due in part to the high incidence of UTIs, but recurrent infections also play an important role. The rate of recurrence of infection within six months exceeds 25% despite adequate management with antibiotics [4].

UTIs are typically treatable with antibiotic therapy, including first-line treatments such as fosfomycin, nitrofurantoin, and trimethoprim/sulfamethoxazole (TMP/SMX) [5]. Other treatments, including fluoroquinolones, are available but have more significant side effects than first-line treatments [6]. However, the rates of bacterial resistance to antibiotics are rising, potentially limiting treatment options in the future [7]. Many species of bacteria that commonly cause UTIs, including *Escherichia coli*, have shown resistance to multiple common antibiotic treatments. For example, the rates of resistance to TMP/SMX exceed 20% in some regions, preventing their use [8]. Nitrofurantoin and fosfomycin exhibit lower rates of resistance [9]. Increasing rates of multi-drug resistant UTI pathogens makes treatment more challenging and are especially concerning considering antibiotics are currently the only approved treatment for a UTI [10].

In addition to issues surrounding mounting resistance, antibiotic use frequently comes with unwanted side effects. Antibiotics can cause disturbances to the body’s normal microbiome, specifically in the genitourinary and gastrointestinal tracts. These disturbances can result in candidiasis, which requires additional antimycotic therapy to resolve [11]. Antibiotics also take time to fully resolve the infection, meaning that symptoms of pain and discomfort can linger even after treatment has begun. Colgan and Williams noted that patients only begin to experience relief from UTI symptoms after 36 h [12]. Patients may need to take additional medications to manage these symptoms. Taken together with the threat of antibiotic resistance, these findings illustrate the need for additional, non-antibiotic-based strategies for the management of UTIs.

One strategy of interest for the management of this painful inflammatory condition is the modulation of the endocannabinoid system (ECS). The ECS plays an important role in the maintenance of physiological homeostasis, with a variety of effects across the entire body [13,14]. Major receptors in the system include the G-protein-coupled receptors cannabinoid type 1 receptor (CB_1_R) and cannabinoid type 2 receptor (CB_2_R) [13]. CB_1_R agonism has anti-inflammatory and anti-nociceptive effects, but its prominence within the central nervous system (CNS) also results in psychoactive effects that make it an undesirable target for pain relief. In contrast, CB_2_R is primarily found outside of the CNS, and receptor activation does not produce psychoactive effects. Within the bladder, CB_2_R is expressed in the both the mucosa and the detrusor and is known to be upregulated in response to inflammation [15,16]. We have recently demonstrated the beneficial effects of CB_2_R activation by the agonist beta-caryophyllene (beta-C) in a murine model of interstitial cystitis [17]. While interstitial cystitis is not infectious by definition, there is significant overlap between the symptoms of interstitial cystitis and UTI, with both conditions resulting in significant pain and inflammation.

Beta-C is a non-psychoactive sesquiterpene that occurs naturally in a variety of plants, including cannabis, cloves, and black pepper [18]. A recent systematic review of the literature identified a beta-C content ≥ 10% in approximately 300 species across 51 different families [19]. Beta-C content reaches approximately 60% in certain species of fir tree (*Abies*) [20]. Beta-C is a generally recognized as safe (GRAS), meaning that experts have deemed it safe for human consumption as it has a favorable toxicological profile [21]. While not a classical cannabinoid compound, beta-C acts though the endocannabinoid system by selectively and specifically binding CB_2_R to produce anti-inflammatory effects [22]. In addition to its CB_2_R-mediated effects, beta-C also exerts local anesthetic effects [23]. Its anti-bacterial activity has also been identified. Beta-C has shown anti-bacterial activity against multiple species of bacteria, including *E. coli* [22,23]. When taken in combination with the anti-inflammatory and associated with CB_2_R agonism, these findings indicate that beta-C may represent a valuable addition to traditional antibiotic-based strategies for managing UTIs. As such, this study aims to investigate the effects of beta-C treatment in a murine model of UTI.

## 2. Results

We observed a significant but variable increase in the urinary bacterial burden six hours post inoculation (Figure 1A) which was not significantly reduced by co-treatment with beta-C, fosfomycin, or combination therapy. In subsequent experiments, additional urinary samples were collected six hours post inoculation, just prior to treatment with beta-C to minimize the variability observed in preliminary work. The animals were followed for 24 and 72 h to monitor the progress of infection. The urinary bacterial burden remained high 24 h post inoculation and typically resolved by 72 h regardless of treatment status (Figure 1B,C). After 24 h, beta-C treatment resulted in a significant reduction in urinary bacterial burden (Figure 1A). There was no significant difference between treatment with 100 mg/kg beta-C and a low dose of fosfomycin (10 mg/kg), and no additional benefit from combination therapy was identified with respect to the bacterial burden. The anti-bacterial effects of beta-C were not abrogated by a blockade of the CB_2_R with antagonist AM630 (Figure 1B).

Adhesion and invasion into bladder epithelial cells represent critical points in UTI pathogenesis. We detected low levels of bacterial burden within bladder tissues after 24 h, suggesting some intracellular infiltration occurs at this timepoint (Figure 2A). Treatment with either beta-C or low dose fosfomycin resulted in a significant reduction in tissue bacterial burden, below the limit of detection of the enumeration method. This effect persisted with the administration of AM630. The low levels of infection within tissues are supported by the limited tissue damage observed via histological staining (Appendix A). Immune cells were primarily localized to blood vessels in sham animals (Appendix A), with some localized infiltration in infected animals (Appendix A).

Following successful bladder infection, UTIs may progress into other tissues. Preliminary data indicates that infection spreads sporadically into the kidney and spleen six hours post inoculation but does not progress to systemic infection of the blood (Figure 2B–D). The bacterial load seen at this early timepoint within some kidneys and spleens appears to resolve spontaneously by the 24 h timepoint (Figure 2B,C), suggesting that the infection generally remains localized to the lower urinary tract. Blood remained sterile at all timepoints (Figure 2D). Levels of pro-inflammatory cytokines within the plasma six hours post inoculation are also indicative of localized infection (Appendix A). Within the tissue, elevated IL-6 was observed (Appendix A).

As UTIs frequently result in painful symptoms that persist beyond the administration of antibiotic therapy, we also evaluated the analgesic effects of beta-C treatment. After UTI induction, the animals exhibited significantly increased signs of non-evoked pain, as measured by behavioral scoring (Appendix A). Behavioral changes persisted from 6 to 24 h post induction and were significantly reduced by treatment with beta-C, both alone and in combination with fosfomycin (Figure 3A,B). In addition, signs of non-evoked pain persisted at 72 h post induction, despite the limited urinary bacterial burden observed (Figure 3C). The administration of the antagonist AM630 did not block the anti-nociceptive effects of the beta-C treatment (Figure 3B). We also evaluated the animals’ responsiveness to evoked pain via von Frey esthesiometry (Appendix A). The animals exhibited a significant decrease in their tolerance for applied suprapubic pressure at the 24 h timepoint which did not resolve with beta-C treatment alone (Appendix A). Treatment with beta-C in combination with AM630 significantly reduced this response. No significant changes in pain response were observed at either 6 or 72 h post infection (Appendix A).

Using IVM, we investigated the effect of beta-C treatment on inflammatory changes within the bladder microvasculature. Adherent leukocytes within venules of an infected bladder are shown in Figure 4D. We observed a significant increase in both rolling and adherent leukocytes 24 h after UTI induction, representing an increase in leukocyte activation (Figure 4A,B). Treatment with beta-C reduced levels of adherent but not rolling leukocytes. AM630 administration did not prevent the observed reduction in adherent leukocytes (Figure 4B). Fosfomycin treatment also resulted in a significant decrease in adherent leukocytes. We also assessed FCD as a marker of changes in microvascular function related to inflammation. UTI induction resulted in a significant decrease in FCD, which was restored with beta-C treatment (Figure 3C). AM630 administration partially reduced the effect of beta-C treatment on FCD.

## 3. Discussion

The primary goal of this study was to evaluate the sesquiterpene beta-C for the management of UTIs, both alone and in combination with antibiotic treatment. While UTIs are generally treatable with standard antibiotic therapies, these therapies do not adequately manage pain or inflammation, can have unpleasant side effects, and may have limited efficacy against antibiotic-resistant pathogens. Based on the anti-inflammatory and anti-nociceptive profile of beta-C and the emerging data suggesting anti-bacterial activity, we hypothesized that beta-C administration would be a promising approach for the management of UTIs.

We first sought to evaluate the anti-bacterial effects of beta-C in a murine model of UTI. The intravesical administration of UPEC resulted in UTI which persisted from 6 to 24 h and resulted in limited systemic spread. Using an optimized timepoint of 24 h, we identified anti-bacterial effects in both the urine and bladder tissues of female BALB/c mice. A significant reduction in the bacterial burden was observed in both the urine and bladder tissues, suggesting that beta-C has anti-bacterial effects in this setting. Several studies have previously demonstrated similar effects in vitro. Dahham et al. demonstrated anti-bacterial activity against six bacterial strains, including *E. coli* [24]. Similarly, Neta et al. demonstrated anti-bacterial activity against *E. coli* in infected murine hepatoma cells [25]. Several other studies have also shown anti-bacterial affects against a variety of pathogens, but the mechanism of activity is yet to be described [26,27]. Importantly, we observed no significant differences in either urinary or bladder tissue bacterial burden between treatment with either beta-C or low-dose fosfomycin, which suggests beta-C may have similar anti-bacterial efficacy to fosfomycin at the selected dosage.

We next sought to evaluate the anti-nociceptive effects of beta-C in our model of murine UTI. We observed a significant increase in signs of non-evoked pain 6 h post-inoculation, which persisted from 24 to 72 h. Significant evoked pain responses were only observed at the 24 h timepoint. The pain levels were highly variable despite the exclusion of mice without robust infections at the 24 and 72 h timepoints. A similar study by Rudick et al. determined that there was no correlation between pelvic pain and bladder colonization, but there are no data available with respect to non-evoked pain [28]. Beta-C treatment, either alone or in combination with fosfomycin, was able to significantly reduce the levels of non-evoked pain at both the 6 and 24 h timepoints. The administration of CB_2_R antagonist AM630 did not abrogate this effect, which is likely attributable to the local anesthetic effects associated with the beta-C treatment [23].

Although UTIs are frequently associated with painful symptoms, the mechanisms responsible remain unclear. This is illustrated by the lack of painful symptoms present in cases of asymptomatic bacteriuria, defined as the presence of more than 10^5^ CFU/mL within the urine without typical symptoms of infection [29]. A recent study by Rosen and Klumpp suggests that early pain responses are related to interactions between LPS and TLR4 and may be separated from inflammatory pathways [30]. The authors further determined that TRPV1 and the MCP-1/CCR2 axis are involved [31]. These findings suggest that the majority of UTI pain is mediated by receptors other than CB_2_R and may explain the variable responses to beta-C treatment that we observed in this study. There is currently no consensus regarding the effects of beta-C on TRPV1, but evidence suggests that beta-C inhibits MCP-1 secretion [32,33,34]. This suggests beta-C may also represent a valuable strategy for managing pain associated with either TRPV1 or the MCP-1/CCR2 axis.

The onset of UTI is associated with a significant inflammatory response, characterized by the production of pro-inflammatory meditators by both urothelial cells and immune cells. Major cytokines produced during UTI include TNF-α, IL-1β, IL-6, IL-8, and IFN-γ [35]. We did not observe significant increases in tissue levels of the majority of cytokines investigated at either the 6 h or 24 h timepoints. This may be related to the attenuation of the immune response by the UPEC strain CFT073, which interferes with both TLRs and the NLRP3 inflammasome via the expression of Toll/IL-1-receptor-containing protein C (TcpC) [36,37]. A significant but low level of IL-6 was observed at the early timepoint, which correlates with the partial suppression observed by Yadav et al. [38].

Despite relatively low levels of cytokine expression, including neutrophil chemoattractant CXCL2, we observed a significant influx of leukocytes into the bladder microvasculature, with IVM showing significant increases in the number of both rolling and adherent leukocytes. Previous studies have also shown the rapid recruitment of neutrophils into inflamed bladder tissue, with levels peaking 24 h post infection [39,40]. Treatment with beta-C resulted in a significant decrease in the number of adherent leukocytes, which aligns with our previous study investigating the anti-inflammatory effects of beta-C in a sterile model of bladder inflammation [17]. CB_2_R activation has also had similar effects in other inflamed tissues, such as the retina, brain, and small intestine [41,42,43]. These effects could be reversed by the administration of the pharmacological CB_2_R antagonist AM630, which contrasts with what we observed in our study. Levels of rolling leukocytes were unaffected by beta-C treatment, either alone or in combination with AM630. Other studies have shown mixed results with respect to the effects of CB_2_R activation on leukocyte rolling, indicating that more research is needed to fully understand the effects of beta-C in this context [44,45,46]. Importantly, beta-C treatment had similar anti-inflammatory effects when compared to fosfomycin therapy alone, which is known to exert some immunosuppressive effects on leukocytes [47].

Microcirculatory changes in response to inflammation have been described in the bladder, including a reduction in functionally perfused capillaries and decreased red blood cell velocity [17,48]. To our knowledge, this is the first study to describe these changes in the context of UTI. We observed a significant decrease in FCD 24 h post UTI induction, which was restored to baseline levels with beta-C treatment. This mirrors findings seen in a sterile model of bladder inflammation in which CB_2_R activation by either beta-C or agonist HU-308 resulted in significant improvements in FCD [17]. Notably, CB_2_R receptor blockade with AM630 did not completely abrogate the improvements in capillary perfusion seen with beta-C treatment. While this finding suggests some role for CB_2_R activation in the modulation of microcirculatory perfusion, it also suggests the involvement of other pathways. A wide variety of factors are known to affect microcirculatory perfusion, including altered blood flow regulation, changes in capillary permeability, and the activation of the coagulation cascade [49,50]. As none of our treatments were able to fully restore microvascular perfusion during UTI regardless of reductions in the bacterial burden/inflammation, more research into the mechanisms at play is merited.

## 4. Materials and Methods

### 4.1. Bacterial Strain and Culture Conditions

The uropathogenic *Escherichia coli* (UPEC) strain CFT073 (American Type Culture Collection 700928, Cedar Lane, Burlington, ON, Canada), isolated from a female pyelonephritis patient, was used in this study [51]. Rifampicin resistance was induced via repeated subculturing in sublethal concentrations of rifampicin as previously described [52]. Bacteria were routinely grown on nutrient agar plates at 37 °C with rifampicin as required (50 μg/mL in dimethyl sulfoxide). Prior to infection, UPEC were grown overnight in nutrient broth (with shaking at 37 °C), and then subcultured into fresh broth to encourage the expression of type 1 fimbriae.

### 4.2. Murine Model of Lower UTI

Female BALB/C mice (8–12 weeks old, 18–25 g) were purchased from Charles River Laboratories International Inc. (Wilmington, MS, USA) and housed in ventilated cage racks at the Carleton Animal Care Facility, Faculty of Medicine, Dalhousie University, Halifax, NS, Canada. The animals were provided with a standard diet of rodent chow with water access ad libitum were and kept on a 12 h light/dark cycle. All experimental protocols were performed following the guidelines set forth by the Canadian Council on Animal Care and approved by the Dalhousie University Committee on Laboratory Animals.

The mouse infection studies were performed as previously described by Hannan and Hunstad, with some modifications [53]. In brief, the animals were anesthetized using isoflurane, and the bladder was manually voided. A sterile catheter (P10 polyethylene tubing) was inserted transurethrally, and 50 µL of UPEC (1–2 × 10^8^ CFU/mL in D-PBS) was slowly instilled into the bladder to minimize the chance of vesicoureteral reflux. The animals received intraperitoneal injections of beta-caryophyllene ((≥80%, ≤19% C15H24 hydrocarbons, Sigma-Aldrich, St. Louis, MO, USA; 100 mg/kg, 20 mg/mL in an olive oil vehicle), fosfomycin (Verity Pharmaceuticals, Mississauga, ON, Canada; 10 mg/kg, 33.3 mg/mL in saline) or a combination therapy of beta-C and fosfomycin. Beta-C doses were based on in vivo toxicity studies which indicated that the lethal dose in murine models exceeds 2000 mg/kg [54]. For preliminary studies, the animals were treated immediately following the induction of UTI and then evaluated after six hours. To enhance clinical relevance, the animals in extended timepoints received treatment 6 h after induction and were then evaluated after either 24 or 72 h post-induction. For extended timepoints, animals were excluded from the analysis if the urinary bacterial burden fell below 10^3^ CFU/mL. All animals were humanely sacrificed using approved protocols.

Antagonist studies were conducted at the 24 h timepoint to determine the effects of a CB_2_R antagonist on the efficacy of beta-C treatment. AM630 (Millipore Sigma, Burlington, MA, USA; 2.5 mg/kg) was administered via intraperitoneal injection 30 min prior to beta-C treatment. Mice were evaluated as previously described 24 h post induction.

### 4.3. Pain and Behavior Assessment

Electronic von Frey esthesiometry (IITC Inc. Life Science 2390 series, Woodland Hills, CA, USA) was performed to assess changes in pain tolerance. The suprapubic region was probed to measure the amount of force tolerated (in grams) before the animal reacted (e.g., withdrawal). Prior to assessment, animals were acclimatized to the von Frey apparatus on each of the two days prior to experimentation. Prior to UTI induction, the animals were allowed to acclimate in the quiet, dim procedure room for one hour prior to assessment. The animals were then moved to a plexiglass enclosure with a mesh floor (IITC Life Sciences, Woodland Hills, CA, USA) and allowed to acclimatize for 15 min in the presence of the observer. The esthesiometer, with the rigid tip attached, was slowly applied to the suprapubic region until withdrawal occurred. Five values were recorded, allowing at least 30 s between measurements, forming a baseline for pain tolerance. This procedure was repeated at either 6, 24 or 72 h post UTI induction.

Changes in behavior were assessed using a behavioral scoring system adapted from Boucher et al. [55]. A baseline score was recorded prior to the von Frey assessment and UTI induction. Posture, motor activity, and eye opening were assessed on a scale of 1–10 for a maximum score of 30. Full assessment criteria are available in the Appendix A. This procedure was repeated at either 6, 24 or 72 h post UTI induction.

### 4.4. Intravital Microscopy

Intravital microscopy (IVM) of the bladder was used to assess changes in immune activation, as evidenced by leukocyte rolling and adhesion and changes in the capillary perfusion of the bladder wall. IVM was performed 24 h after UTI induction using an epifluorescent microscope (Leica, DM LM, Wetzlar, Germany), set to 20× magnification, and a light source (LEG EBQ 100, Jena, Germany). Briefly, the animals were anesthetized to surgical depth with sodium pentobarbital (65 mg/kg; Ceva Sante Animale, Montreal, QC, Canada) and maintained with repeated administration of 5 mg/kg sodium pentobarbital. Just prior to IVM, the animals were injected via the tail vein with Rhodamine 6G (1.5 mL/kg, 0.75 mg/kg body weight, Sigma-Aldrich, Oakville, ON, Canada) and fluorescein isothiocyanate (FITC)-albumin (1 mL/kg, 50 mg/kg, Sigma-Aldrich, Oakville, ON, Canada). Rhodamine 6G allows for the visualization of leukocytes, while FITC-albumin is used to visualize functional capillary density by illuminating the capillary beds of the bladder. The bladder was inflated with sterile saline to improve visualization. Leukocytes were visualized using green light, and six to eight visual fields containing bladder venules were randomly selected and recorded for 30 s. Capillary blood flow was then examined using FITC-albumin under blue light. Again, six to eight randomly selected visual fields with capillaries were recorded for 30 s.

### 4.5. Bacterial Enumeration

Tissue and fluid samples were collected at the experimental endpoint to quantify the bacterial burden in various organs. Urine and plasma were collected, serially diluted, and plated onto nutrient agar plates containing rifampicin. Tissues (bladder, kidneys, and spleen) were aseptically removed and homogenized prior to serial dilution and plating. Plates were incubated at 37 °C, and colony-forming units were counted after 16–24 h.

### 4.6. Analysis of Bladder Tissue Cytokines

Bladder tissue samples were collected at the experimental endpoint for cytokine analysis. The bladder was bisected along the transverse plane, and the upper dome of the bladder was flash frozen in liquid nitrogen and then stored at −80 °C. Protein concentration was determined via bicinchoninic acid assay (BCA), according to the manufacturer’s instructions (Rapid Gold BCA, Protein Assay, Thermo Fisher Scientific, Waltham, MA, USA). A custom-designed mouse cytokine 10-plex kit (R&D Systems, Minneapolis, MN, USA) was used to determine the concentrations of the following cytokines in tissue homogenates: intercellular adhesion molecule 1 (ICAM-1), interleukin 1 (IL-1), interleukin 10 (IL-10), interleukin-6 (IL-6), chemokine CXC ligand 1 (CXCL1), chemokine CXC ligand 2 (CXCL2), P-selectin, interferon-gamma (IFN-γ), LIX recombinant mouse CXC motif chemokine 5 (LIX), and tumor necrosis factor alpha (TNF-α). Samples were prepared according to the manufacturer’s protocol and read using a Bio-Plex 200 Analyzer with Bio-Plex Manager software (Bio-Rad, Mississauga, ON, Canada).

### 4.7. Bladder Histopathology

The lower portion of the bisected bladder was fixed in 10% neutral buffered formalin and then stored in 70% ethanol prior to processing to prevent the overfixation of tissues. Tissue samples were processed and embedded in paraffin, and then 5 μm sections were cut and transferred onto glass slides for staining. The samples were stained using a standard protocol for hematoxylin and eosin stain (H&E). The bladder samples were scored used a bladder inflammation grading scale adapted from Hopkins et al. [56]. In brief, the tissues were evaluated based on the presence and localization of immune cells and tissue edema/necrosis.

### 4.8. Statistical Analysis

Data are expressed as the mean ± standard deviation (SD). Statistical analyses were conducted using GraphPad Prism 6.0 (GraphPad Software Inc., La Jolla, CA, USA). The data were evaluated for normality using the Kolmogorov–Smirnoff test, and outliers were removed using Tukey’s method. Parametric data were then assessed via a one-way ANOVA with Tukey’s multiple comparisons test. Non-parametric data were assessed using the Kruskal–Wallis test with Dunn’s multiple comparisons test. Differences of *p* < 0.05 were considered statistically significant.

## 5. Conclusions

Our study investigated the effects of CB_2_R agonist beta-C on infection, pain, and inflammation during UTI. Beta-C treatment resulted in a significant reduction in bacterial burden within both the urine and the bladder tissue 24 h post infection. We also observed a significant reduction in markers of pain and inflammation, including behavioral scoring, leukocyte adhesion, and FCD. Further research is needed to fully understand the effects of beta-C in the context of UTI, including the mechanisms of both anti-bacterial and anti-nociceptive effects. These results provide evidence supporting the utility of beta-C as a new adjunct therapy for the management of UTIs.

## Figures and Tables

**Figure 1 molecules-28-04144-f001:**
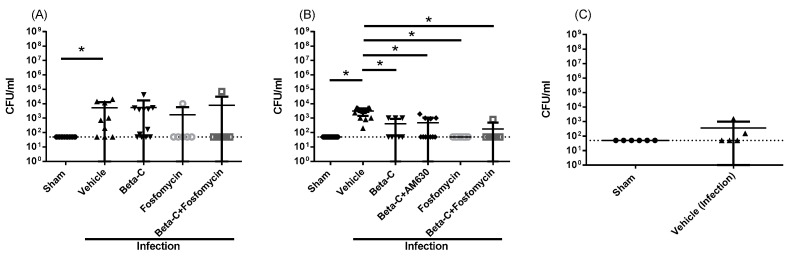
Bacterial burden in the urine of mice (**A**) 6 hours post inoculation, (**B**) 24 h post inoculation, and (**C**) 72 h post inoculation. Animals at the 6 h timepoint were pre-treated at T = 0, while animals at the 24 h timepoint were treated at T = 6 h post inoculation. The dotted line represents the limit of detection (LOD) for the culture method in colony-forming units (CFU) per milliliter (50 CFU/mL). Data presented as mean ± SD, * *p* < 0.05.

**Figure 2 molecules-28-04144-f002:**
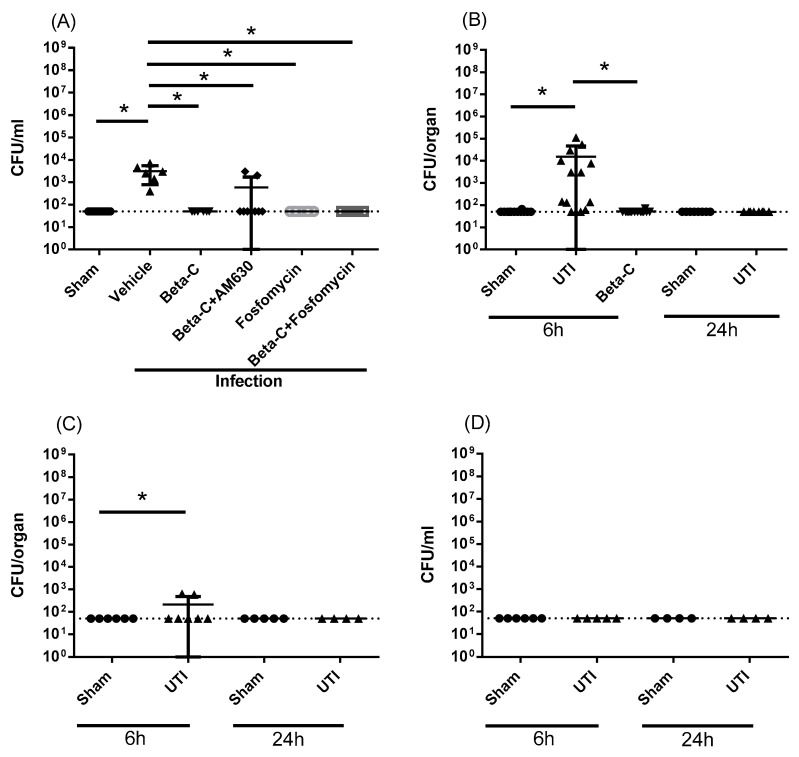
Infection parameters as assessed in mice 6, 24 and 72 h post induction of UTI. Animals at the 6 h timepoint were pre-treated at T = 0, while animals at the 24 h timepoint were treated at T = 6 h post inoculation. Bacterial burden was assessed in the bladder at 24 h (**A**), and within the kidneys at 6 and 24 h (**B**). Systematic parameters were also assessed, including spread to spleen (**C**) and blood (**D**) at 6 and 24 h. Data presented as mean ± SD, * *p* < 0.05.

**Figure 3 molecules-28-04144-f003:**
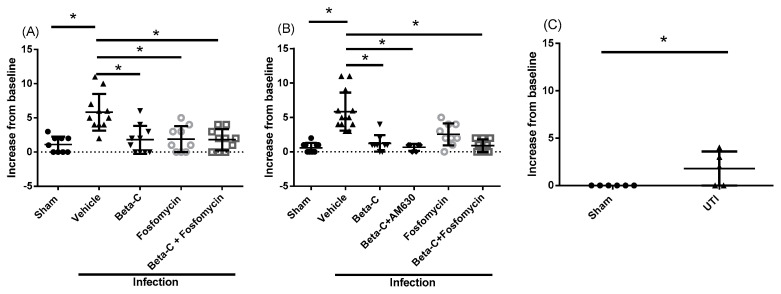
Behavioral changes in infected mice. Changes in behavioral score are shown for mice (**A**) 6 h, (**B**) 24 h, and (**C**) 72 h post inoculation. Animals at the 6 h timepoint were pre-treated at T = 0, while animals at the 24 and 72 h timepoints were treated at T = 6 h post inoculation. Data presented as mean ± SD, * *p* < 0.05.

**Figure 4 molecules-28-04144-f004:**
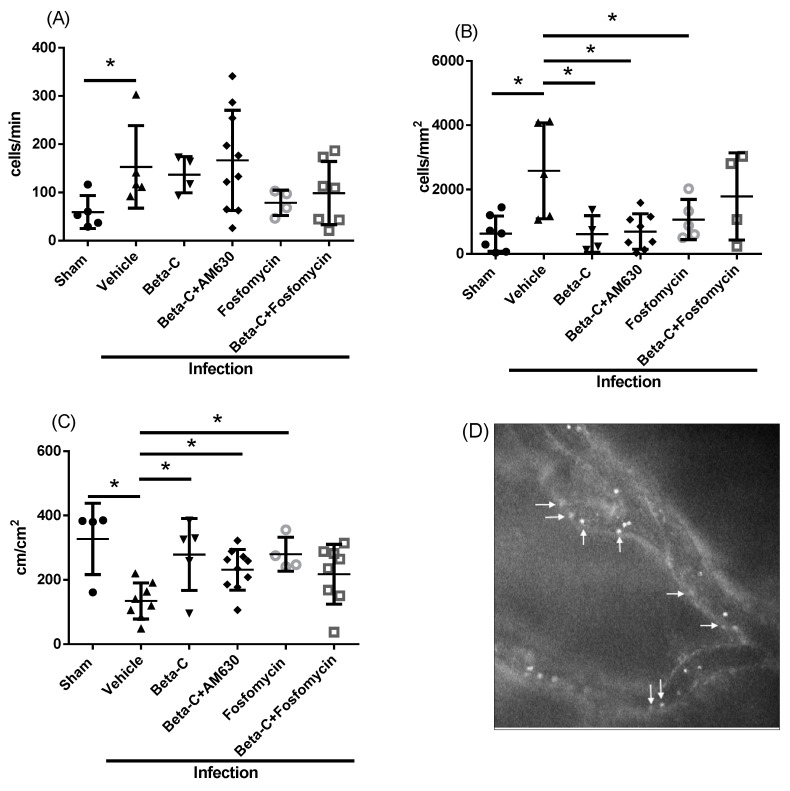
Changes in the bladder microcirculation as assessed in mice 24 h after the induction of a UTI. Using IVM, levels of rolling (**A**) and adhering (**B**) leukocytes were assessed. Functional capillary density (FCD; (**C**)) was also evaluated. Animals were treated at T = 6 h post inoculation and underwent IVM at T = 24 h. Adherent leukocytes are shown within the bladder venules at a magnification of 200× (**D**), as indicated by white arrows. Data presented as mean ± SD, * *p* < 0.05.

## Data Availability

Not applicable.

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
