# Peer review of "Antibacterial and Analgesic Properties of Beta-Caryophyllene in a Murine Urinary Tract Infection Model"

_molecules, 2023, doi:10.3390/molecules28104144_

Round 1

Reviewer 1 Report

The manuscript presented by the authors deals with evaluating beta-caryophyllene in a murine urinary tract infection model. During their research, the authors propose using this natural product as a single molecule or in conjunction with other antibiotics. Although the article has merit in terms of the methodology employed, the reviewer has some concerns in the methodology section that the authors should clarify before recommending acceptance of the manuscript in Molecules:

1) The results in Figure 1 suggest that there is no statistically significant difference in the results obtained with either fosfomycin or beta-caryophyllene, and even some phenomenon of synergism is not discernible. The reviewer doubts whether this is attributable to the high dose of beta-caryophyllene used only allows us to see the already known antimicrobial effect of this compound. Why was such a high dose chosen?

2) As for the other tests performed, it is curious that the antagonistic effects of AM630 are not observed, which could indicate that beta-caryophyllene has a weak agonist effect even at such a high dose or that the dose of the terpene was so high that the antagonistic effects of AM630 are not seen. Studies at other doses may be necessary to establish the role of the activation of the endocannabinoid pathway in inflammation.

Author Response

 Dear Reviewer,

We appreciate the valuable feedback from the reviewers and have provided point by point responses, as outlined below. Changes to the original manuscript are highlighted in yellow for clarity.

  1. The results in Figure 1 suggest that there is no statistically significant difference in the results obtained with either fosfomycin or beta-caryophyllene, and even some phenomenon of synergism is not discernible. The reviewer doubts whether this is attributable to the high dose of beta-caryophyllene used only allows us to see the already known antimicrobial effect of this compound. Why was such a high dose chosen?

The design for this experiment was based on doses used in previous manuscripts. In particular, our group has previously used the 100mg/kg dose in an experimental model of interstitial cystitis (Berger et al., 2019).

  1. As for the other tests performed, it is curious that the antagonistic effects of AM630 are not observed, which could indicate that beta-caryophyllene has a weak agonist effect even at such a high dose or that the dose of the terpene was so high that the antagonistic effects of AM630 are not seen. Studies at other doses may be necessary to establish the role of the activation of the endocannabinoid pathway in inflammation.

We agree with the reviewer’s comment, and are currently planning some in vitro experiments to investigate this further. Of note, in the paper by Berger et al. (2019), experiments with CB2R knockout mice showed that the anti-inflammatory effects of beta-caryophyllene were lost with the absence of the receptor. We observed this with respect to both leukocyte adhesion and capillary perfusion.

We appreciate your time and commentary and are available to provide any further information you may require.

Yours sincerely,

Dr. Christian Lehmann

Reviewer 2 Report

The manuscript described the Antibacterial and analgesic properties of beta-caryophyllene in  a murine urinary tract infection model. In the manuscript, some corrections are required before publications.

Comment 1:   What is the reason to store the bisected bladder in 70% ethanol?

Comment 2:   Combine the figure 1 and Supplemental figure 1(A) and make one figure for the manuscript.

Comment 3:   Combine the figure 2 and Supplemental figure 1(B, C, D) and make one figure for the manuscript.

Comment 4:   Author had performed Bladder histopathology, but they did not supply the images from where authors calculated the cellular damages score. Author should supply the images of bladder and also describe the process for damage score.

Comment 5:   Authors are mentioned in material method section that IVM was performed 24 hours. But in figure 4 (D)mentioned T=6h post inoculation. Please explain.

Comment 6:   Author should supply colourful image of IVM, as Rhodamine 6G and fluorescein isothiocyanate (FITC)-albumin were used.

Author Response

Dear Reviewer,

We appreciate the valuable feedback from the reviewers and have provided point by point responses, as outlined below. Changes to the original manuscript are highlighted in yellow for clarity.

  1. What is the reason to store the bisected bladder in 70% ethanol?
    Bladders are stored in 70% ethanol while awaiting embedding and processing to prevent tissue overfixation.

  1. Combine the figure 1 and Supplemental figure 1(A) and make one figure for the manuscript.
    This has been completed.

  1. Combine the figure 2 and Supplemental figure 1(B, C, D) and make one figure for the manuscript.
    This has been completed.

  1. Author had performed Bladder histopathology, but they did not supply the images from where authors calculated the cellular damages score. Author should supply the images of bladder and also describe the process for damage score.
    We have added an additional figure to clearly show the tissue damages. The method section has also been updated to better describe the scoring process.

  1. Authors are mentioned in material method section that IVM was performed 24 hours. But in figure 4 (D)mentioned T=6h post inoculation. Please explain.
    T=6h refers to the time of treatment post inoculation. We have adjusted the figure caption to more clearly reflect this. Author should supply colourful image of IVM, as Rhodamine 6G and fluorescein isothiocyanate (FITC)-albumin were used.

Our video output does not produce colored images.

We appreciate your time and commentary and are available to provide any further information you may require.

Yours sincerely,

Dr. Christian Lehmann

Reviewer 3 Report

1.       Caryophyllene is one of the common sesquiterpene hydrocarbons. It is found in several tens of thousands of plant species. Therefore, it is wrong to limit its prevalence three types of row materials. The nature of Canada is rich in unique species of coniferous plants, the essential oil of which is a source of caryophyllene as well as other essential oil plants.

2.       Caryophyllene was first isolated from resin of Pinus martima Poir. and then found in other types of coniferous plants. For example, in resin of Abies broxteata Don., the content of caryophyllene is about 60%. Therefore, it is worth including publications in the list of references Dupont Y., Dulou R., Naffa P. Bull. Soc. Chim. France. 1948, 9-10, 990-994. SmedmanL.A.,  Snajberk K., Zavarin E., Mon T. R. Phytochemistry. 1969, 8, 8, 1471-1479.

3.       It is not worth quoting article 18, since its name misleads readers, classifying caryophyllene as a cannabinoid.

4.       In the Materials and Methods section, the purity of caryophyllene taken for the experiment is 80%. The composition of the main impurities should be indicated, since they can demonstrate the observed effect.

5.       In the list of references, the names of bacteria and plants should be written in italic, as in the main text.

6.       The authors use a high dose of caryophyllene, bordering on lethal. It should be noted that the dose is selected taking into account acute and chronic toxicity tests.

Author Response

Dear Reviewer,

We appreciate the valuable feedback from the reviewers and have provided point by point responses, as outlined below. Changes to the original manuscript are highlighted in yellow for clarity.

  1. Caryophyllene is one of the common sesquiterpene hydrocarbons. It is found in several tens of thousands of plant species. Therefore, it is wrong to limit its prevalence three types of row materials. The nature of Canada is rich in unique species of coniferous plants, the essential oil of which is a source of caryophyllene as well as other essential oil plants.

We have adjusted the manuscript to reflect the diversity of sources of beta-caryophyllene. We have included a systematic review that identified beta-caryophyllene as a major component (≥10%) in approximately 300 species across 51 families (Maffei, 2020).

  1. Caryophyllene was first isolated from resin of Pinus martima and then found in other types of coniferous plants. For example, in resin of Abies broxteata Don., the content of caryophyllene is about 60%. Therefore, it is worth including publications in the list of references Dupont Y., Dulou R., Naffa P. Bull. Soc. Chim. France. 1948, 9-10, 990-994. SmedmanL.A.,  Snajberk K., Zavarin E., Mon T. R. Phytochemistry. 1969, 8, 8, 1471-1479.

We have integrated the Smedman (1969) reference into the introduction of the manuscript. We have not included the Dupont (1948) as we could only find French versions of the text.

  1. It is not worth quoting article 18, since its name misleads readers, classifying caryophyllene as a cannabinoid.

This misclassification is noted, but we have left the reference in place as additional references also refer to beta-caryophyllene as a dietary cannabinoid (da Silva Oliveira et al., 2018).

  1. In the Materials and Methods section, the purity of caryophyllene taken for the experiment is 80%. The composition of the main impurities should be indicated, since they can demonstrate the observed effect.

Other compounds include C15H24 hydrocarbons. We have updated the methods section to reflect this.

  1. In the list of references, the names of bacteria and plants should be written in italic, as in the main text.

The reference section has been updated.

  1. The authors use a high dose of caryophyllene, bordering on lethal. It should be noted that the dose is selected taking into account acute and chronic toxicity tests.

We have adjusted the methods section to include a reference (da Silva Oliveira et al., 2018), which shows that the lethal dose of beta-caryophyllene in vivo in mice exceeds 2000 mg/kg.

We appreciate your time and commentary and are available to provide any further information you may require.

Yours sincerely,

Dr. Christian Lehmann

Round 2

Reviewer 1 Report

Since the authors have clarified my doubts, the reviewer is in position to recommend the acceptance of the manuscript. However, the reviewer still considers that further experimentation is needed to evaluate whether the dose of 100 mg/kg can be reduced.

Reviewer 2 Report

Thank you for providing me the opportunity to review the manuscript entitled “Antibacterial and analgesic properties of beta-caryophyllene in a murine urinary tract infection model”. The authors have modified the manuscript as per the comments of the reviewers. The manuscript can be accepted in its current form.